# Alzheimer's risk factor FERMT2 promotes the progression of colorectal carcinoma via Wnt/β-catenin signaling pathway and contributes to the negative correlation between Alzheimer and cancer

**Wenzhen Xia**[1], **Zhaoyu Gao**[1,2,3], **Xia Jiang**[1,4], **Lei Jiang**[1,2,3], **Yushi Qin**[1], **Di Zhang**[1],
**Pei Tian**[1], **Wanchang Wang**[1], **Qi Zhang**[1], **Rui Zhang**[1,2,3], **Nan Zhang**[1,2,3],
**Shunjiang Xu**[1,2,3] *

**1** Central Laboratory, The First Hospital of Hebei Medical University, Shijiazhuang, Hebei, China, **2** Hebei International Joint Research Center for Brain Science, Shijiazhuang, Hebei, China, **3** Hebei Key Laboratory of Brain Science and Psychiatric-Psychologic Disease, Shijiazhuang, Hebei, China, **4** Key Laboratory for Colorectal Cancer Precision Diagnosis and Treatment of Hebei Province, Shijiazhuang, Hebei, China

* xushunjiang@hebmu.edu.cn

## Abstract

Increasing evidence from epidemiological studies indicate that Alzheimer's disease (AD) has a negative relationship with the incidence of cancers. Whether the Alzheimer's genetic risk factor, named as fermitin family homolog-2 (FERMT2), plays a pivotal part in the progressive process of colorectal carcinoma (CRC) yet remains unclear. This study revealed that FERMT2 was upregulated in CRC tissues which predicted an unfavorable outcome of CRC using the PrognoScan web tool. FERMT2 was co-expressed with a variety of genes have been linked with CRC occurrence and implicated in the infiltration of immune cell in CRC tissues. Overexpressing FERMT2 promoted CRC progression with upregulation of Wnt/β-catenin signaling. Knockdown of FERMT2 suppressed the cell multiplication, colony formation rate, migration and invasion, along with the epithelial to mesenchymal transition (EMT) with downregulation Wnt/β-catenin proteins in cells of CRC, while overexpressing β-catenin reversed the inhibitory effects of silencing FERMT2 on the migration or invasion of CRC cells. Furthermore, $A\beta_{1-42}$ treated HT22 cells induced downregulation of FERMT2 and inhibited the migration, invasion and EMT in co-cultured CT26 cells through Wnt/β-catenin signaling. Our results revealed that the downregulated FERMT2 gene during AD is prominently activated in CRC, which promotes its progression via Wnt/β-catenin pathway.

## Introduction

Alzheimer's disease (AD) is a prevalent neurodegenerative disease among the elderly population worldwide. The main pathological lesions in AD have been characterized as the extracellular senile plaques and intracellular neurofibrillary tangles [1]. Cancers, another kind of

**Data Availability Statement:** All relevant data are within the paper and its Supporting Information files.

**Funding:** This work was supported by grants from Hebei Provincial Natural Science Foundation (H2018206358, H2019206565 and H2020206105) and The Science and Technology project of the People's Livelihood in Hebei Province (20377707D). The funders had no role in study design, data collection and analysis, decision to publish, or preparation of the manuscript.

**Competing interests:** The authors have declared that no competing interests exist.

aging-related disorders, are characterized by uncontrolled cell proliferation, invasion and migration. Interestingly, more and more epidemiological studies consistently indicate that there is a negative association between AD and the occurrence of cancers [2–11]. As a result, patients with a diagnose of cancer have a low incidence of AD, whereas patients with AD display a lower risk of cancer attack in the later years [12]. Several studies have shown that the frequent type of cancers negatively associated with AD is colorectal carcinoma (CRC) [3, 5, 6, 13]. However, the mechanisms underlying the negative association between AD and CRC are still unknown.

Fermitin family homolog-2 (FERMT2) is a member of focal adhesion protein that regulates lots of physiological processes including cardiac and skeletal muscle development [14–16], and embryonic development by enhancing integrin activation [15, 17]. Previously several genetic variants have been confirmed to be related with the AD risk [18, 19]. As a genetic risk factor of AD, FERMT2 was identified to be implicated in amyloid precursor protein (APP) metabolism and downregulated FERMT2 promoted Aβ peptide production through facilitating the mature APP recycling and elevating its levels at the cell surface [20]. Recent research has shown that FERMT2 regulates the growth of axons, connectivity of synapses, and long-term potentiation in an APP-dependent pattern [21]. On the other hand, FERMT2 has been linked to tumor pathogenesis and is highly expressed in multiple kinds of cancer, including gastric cancer, breast cancer, and glioblastoma multiforme [22–27]. However, the specific role of FERMT2 in CRC and how the FERMT2 participates in CRC invasion and metastasis remain unknown. Particularly, whether FERMT2 is involved in the reverse relationship between AD and CRC remains unidentified.

In the present study, the expression and function of FERMT2 in CRC progression were analyzed using public databases. The genomic alterations, mutations and target proteins of FERMT2 during development of CRC, as well as its roles in tumor immunization were predicted by bioinformatics analysis. Meanwhile, the implications of FERMT2 in the progression of CRC and its underlying mechanisms were investigated using cell lines. Finally, the interactive effects between co-cultured AD model cells and CRC cells on the migration, invasion, EMT and the proteins linked to Wnt/β-catenin signaling were explored using CT26 cells.

## Materials and methods

### Data source and analysis of gene expression

Data on gene expression were obtained from GEO (Gene Expression Omnibus). As an online database of cancers, Oncomine includes the genes and pathways have been discovered through gene expression microarray assay (http://www.oncomine.org) [28]. In Oncomine, FERMT2 mRNA expression in CRC and control tissues were analyzed and the values were determined according thresholds: fold change >1.5 or gene ranking ($P < 0.05$).

### Survival analyses

The prognostic value of individual genes was analyzed using the PrognoScan meta-analysis database (version 4.5) [29]. During the screening process, we set the parameters as: 1) Set "Cancer Type" as "colon cancer"; 2) Set "THERSHOLD" as "$P < 0.05$". The analysis from PrognoScan was visualized in R with "survminer" and "survival".

### Analysis of co-expression networks

Total of 32 TCGA cancer-related multidimensional datasets were obtained from LinkedOmics database [30]. The co-expression genes of FERMT2 were analyzed using Pearson's correlation

coefficient. The gene set enrichment analysis (GSEA) was performed using LinkedOmics database. The analysis items including Gene Ontology biological process (GO_BP), KEGG pathways, and enrichment of kinase, miRNA and transcription factor targets. The rank criterion was set as False Discovery Rate (FDR) < 0.05.

### GEPIA2 database analysis

The heatmaps of top co-expression genes together with the survival curves of top kinase regulators were plotted using the Gene Expression Profiling Interactive Analysis (GEPIA2) [31].

### Analysis of immune infiltration

The Tumor Immune Estimation Resource (TIMER) provides a systematic analysis of tumor infiltrating immune cells (TIICs) from diverse cancer types in TCGA [32]. The correlation between FERMT2 expression level and the amounts of immune infiltrates, including CD4[+] T cells, CD8[+] T cells, B cells, NK cells and macrophages, was evaluated in CRC using TIMER.

### cBioPortal database analysis and mutation analysis

The multidimensional tumor genomics datasets were provided in the cBio Cancer Genomics Portal [33]. First step, we analyzed all somatic mutation data of patients without exclusion. Then we analyzed FERMT2 mutations in CRC. The somatic mutation data and clinical data of patients with CRC were obtained from the TCGA-COAD project in TCGA database. Finally, we visualized the somatic variation data in Mutation Annotation Format (MAF) with the R package of "maftools", which contains many useful analysis methods and visualization modules in tumor genomics [34].

### Cell culture

The human colonic epithelial cell NCM460 was purchased from BNCC (Henan, China) and all human colorectal cancer cell lines, including SW480, SW1116, SW1463, HT29 and LoVo, were kindly given by the Key Laboratory for Colorectal Cancer Precision Diagnosis and Treatment of Hebei province (Shijiazhuang, Hebei, China). The cells were cultured with DMEM medium (Gibco, Grand Island, NY, USA) containing 10% fetal bovine serum (ExCell Bio, Shanghai, China) and 1% penicillin/streptomycin (Solarbio, Beijing, China), which were placed in an incubator with 5% $CO_2$ at 37˚C.

### Cell transfection and regents

The short hairpin RNAs (shRNA) including shRNA negative control (shNC) and FERMT2 shRNA (shFERMT2-1 and shFERMT2-2) were synthesized by VectorBuilder (Guangzhou, China). Attractene Transfection reagents (Qiagen, Shanghai, China) were used for cell transfection in the light of manufacturer's instructions. Briefly, the day before transfection, SW1116 and SW1463 cells ($2 \times 10^5$ cells /well) were seed in six-well plates in DMEM medium with 10% FBS, and then cultured for 1 day under normal growth conditions. When transfection, dilute 1.2 μg DNA dissolved with DMEM medium without FBS or antibiotics to a volume of 100 μl. For co-transfection, 0.6 μg FERMT2 knockdown plasmid and 0.6 μg β-catenin overexpression plasmid were added to medium without FBS or antibiotics to make the final volume as 100 μl. Then, add 4.5 μl Attractene Transfection reagents to the DNA solution and then incubate the samples for 15 min at room temperature to allow complex formation. Next, add the transfection complexes onto the grown cells and incubated at 37˚C with 5% CO2 for 24 h.

## RT-qPCR assay

The total RNA of the tissue samples and cells were extracted using the RNA-easy™ Isolation Reagent (Vazyme, Nanjing, China). Then, 100 ng of total RNA was reverse transcribed into cDNA using GoScript™ Reverse Transcription Mix, Oligo(dT) (Promega, Madison, WI, USA). The relative expression of mRNA was determined by RT-qPCR with the GoTaq® qPCR Master Mix (Promega, Madison, WI, USA). The RT-qPCR was detected using the ABI PRISM® 7500 Real-Time PCR System (ThermoFisher Scientific, Hampton, NH, USA). The sequences of the PCR primers are as following: 5′-AAATGGTCACCGTAGAGTTTGC-3′ as a forward primer and 5′-CTCTCGTTTTGGTCTTTTGCAC-3′ as a reverse primer for FERMT2; 5′-GGTGAAGGTCGGTGTGAACG-3′ as a forward primer and 5′-CTCGCTCC TGGAAGATGGTG-3′ as a reverse primer for glyceraldehyde-3-phosphate dehydrogenase (GAPDH). The results were standardized to the expression of GAPDH. The $2^{-\Delta\Delta CT}$ method was used to calculate the relative fold changes.

## Western blot analysis

We lysed CRC cells in RIPA buffer (Solarbio, Beijing, China), and determined their protein concentrations by a BCA (Bicinchoninic acid) protein quantification kit (Solarbio, Beijing, China). An amount of 100 μg of protein was separated on an 10% SDS-PAGE gel (Bio-Rad, Hercules, CA, USA) and transferred to a PVDF membrane (Bio-Rad, Hercules, CA, USA). It was then blocked with 5% BSA and incubated with primary antibodies specific for FERMT2 (ab74030, Abcam, Cambridge, UK; 1:1000), β-catenin (8480, Cell Signaling Technology, Danvers, MA, USA; 1:1000), cyclin D1 (2922, Cell Signaling Technology, Danvers, MA, USA; 1:1000), E-cadherin (AF0301, Affinity Bioscience, Jiangsu, China;1:1000), N-cadherin (ab76011, Abcam, Cambridge, UK; 1:1000), Vimentin (5741, Cell Signaling Technology, Danvers, MA, USA; 1:1000) and β-actin (Proteintech, Wuhan, China; 1:1000,), followed by incubation at room temperature for 1 h with a DyLight fluorescent dye-conjugated secondary antibody (Abbkine, Wuhan, China; 1:2000). Odyssey CLx Imaging System (LI-COR Biosciences, Lincoln, USA) detected and scanned protein signals.

## Fluorescence-activated Cell Sorting (FACS)

The expression of β1 integrins was examined by FACS (BD Bioscience, San Jose, CA, USA). Briefly, the transfected SW1116 or SW1463 cells ($1\times10^6$ cells) were washed and resuspended in PBS. Then the cells were transferred into 5 mL tubes to incubate with the labelled antibody β1 integrin (FHF0291, 4A Biotech, Beijing, China; 1:30) for at least 15 min in dark at 4˚C. After incubation, the cells were washed twice to remove the unbound primary antibody. Next, the cells were incubated with fluorescent labelled secondary antibody for at least 15 min in dark at 4˚C. After incubation, the labelled cells were washed and resuspended in the stain buffer to detect using the machine.

## Determination of cell viability

The Cell Counting Kit-8 (Zoman, Beijing, China) assay was used to determine the cell proliferation. Briefly, the indicated cells ($1\times10^3$ cells/well) were seeded into 96-well plates. The cells were then incubated for 2 h after CCK-8 solution had been added to each well at 24, 48, 72, and 96 h, respectively. Finally, the microplate reader (Promega, Madison, WI, USA) was adopted to detect the absorbance of 450 nm.

## Wound healing assay

Collective cell migration was determined by wound healing assay. Briefly, the transfected SW1116 or SW1463 cells ($5\times10^5$ cells/well) were harvested and seeded into 6-well plates. After 24 h, a single scratch was performed in the layer of cells using the 200 μL pipette tip each well. Then, the scratched cells were washed out with PBS and the remaining cells were continuously incubated with fresh medium without FBS. Lastly, wound healing status was photographed at 0 h and 24 h using the microscope. The wound closure area was analyzed using the ImageJ software (National Institutes of Health, Bethesda, MD, USA).

## Transwell assay

For transwell migration assays or invasion assays, transfected SW1116 or SW1463 cells with serum-free medium ($2\times10^5$ cells/100 μl) were seeded on uncoated or Matrigel coated upper chambers (Costar, Boston, MA, USA) respectively, while total of 500 μl medium containing 20% FBS was added in the bottom chambers. After incubation for 24 h, the migrated or invaded cells were fixed by 4% paraformaldehyde for 30 min and then stained for 30 min using 0.1% crystal violet. Then, the migrated or invaded cells were visualized and counted in four random fields via the microscope.

## Colony formation test

The clonogenicity of a single cell was detected by colony formation assay. Briefly, transfected SW1116 or SW1463 cells ($1\times10^3$ cells) were placed in a petri dish with 10 mm diameter and incubated at 37°C with 5% CO2 for 14 days and the medium were changed every 7 days. Colony formation was terminated until the colony was visible to the naked eyes. The cells were washed two times with PBS before being fixed by 4% paraformaldehyde and stained using 0.1% crystal violet on the 14th day. The number of colonies was calculated.

## Drug and drug treatment

$A\beta_{1-42}$ peptide (appearance, white powder; purity, >98%) was purchased from ChinaPeptides (catalog No. 04010011827, Wuhan, Hubei, China). $A\beta_{1-42}$ was dissolved with HFIP, and re-suspended by DMSO and sterilized PBS, then diluted to 100 μM and stored at 4°C. Before use, diluted 100 μM $A\beta_{1-42}$ to 10, 20, 40 and 80 μM with DMEM without FBS and penicillin/streptomycin. Thereafter, the different concentrations of $A\beta_{1-42}$ were used to treat HT22 cells for 24 h. Then, the cell viability was estimated by CCK-8 assay.

**Co-culture of AD model cells with CT26 cells.** AD model cells were prepared as above and collected in medium containing 20% FBS and re-seeded at a density of $2 \times 10^5$ cells/well into a 24-well plate and incubated for 24 h. Then the CT26 cells suspended in serum-free medium ($1.5 \times 10^5$ cells/100 μl) were seeded on uncoated or Matrigel coated upper chambers respectively. After incubated for 24 h, the migrated or invaded cells were fixed and stained 30 min, respectively. The stained cells were visualized and counted in four random fields via the microscope. For protein detection, CT26 cells were collected in medium containing 10% FBS and re-plated at a density of $5 \times 10^5$ cells/well into a 6-well plate. 24 h later, HT22 cells or AD model cells with 10% FBS medium ($5 \times 10^5$ cells/1 ml) were seeded on upper chambers (LAB-SELECT, Hangzhou, China) respectively. After co-cultured for 24 h, the CT26 cells were collected and lysed for proteins detection.

### Statistical analysis

All data are presented as means ± SD. We conducted statistical analyses with GraphPad Prism 5.0 software (San Diego, California, USA). The two-tailed Student's t-test was used to estimate the differences between two groups. One-way ANOVA or Two-way ANOVA were used to test the statistical differences among multiple groups. The *P* value less than 0.05 was considered to demonstrate the significant differences statistically. We repeated all the experiments for three times at least.

## Results

### FERMT2 is highly expressed in CRC tissues

To understand the roles of Alzheimer's risk factor FERMT2 in the pathogenesis of CRC, FERMT2 expression levels were firstly analyzed using public database. It is indicated that the expression of FERMT2 at mRNA level in colon carcinoma, colon adenocarcinoma, colorectal carcinoma and colon carcinoma epithelia were increased significantly than that of control tissues from several datasets (Fig 1A–1E and Table 1). These results indicated that the expression of FERMT2 gene was activated in CRC tissues.

### Upregulation of FERMT2 predicts poor survival of patients with CRC

To estimate the correlation between high FERMT2 expression and the outcomes of patients with CRC, the PrognoScan web tool was used to analyze the survival outcomes in the CRC patients with higher or lower FERMT2 expression. Generally, the CRC patients with higher FERMT2 expression had significantly shorter overall survival (OS), disease-free survival (DFS) and disease-specific survival (DSS) compared to those with lower FERMT2 expression in GSE17536 (209209_s_at) cohorts (Fig 2A–2C). Similarly, in the cohort of GSE17536 (209210_s_at), the patients in low-risk group had significantly better OS and DFS than those in the high-risk group (Fig 2D and 2E). Moreover, GSE14333 and GSE17537 cohorts also confirmed the trend that the patients with high FERMT2 expression had a poor survival outcome (Fig 2F–2I). These results suggested that upregulation of FERMT2 predicts poor survival in CRC patients.

### Co-expression networks of FERMT2 in CRC

To further understand the pathological roles of FERMT2 in CRC, the module of "LinkFinder" was used to identify the co-expression mode in CRC cohort through the LinkedOmics database. As shown in Fig 3A, total of 11714 genes (red dots) displayed significant positive correlations, but 8114 genes (green dots) showed significant negative correlations with FERMT2. The top 50 genes correlated with FERMT2 positively or negatively were exhibited in the heat map (Fig 3B and 3C). Moreover, the detailed description of the genes co-expressed with FERMT2 were listed in S1 Table.

The results of enrichment analysis on Gene Ontology (GO) terms suggested that FERMT2 co-expressed genes primarily participate in promotion of phagosome maturation. In contrast, the histone mRNA metabolic process, protein transmembrane transport, RNA capping, and some other processes are inhibited by the FERMT2 co-expressed genes (Fig 3D). The analysis of KEGG pathway indicated the genes co-expressed with FERMT2 are primarily involved in the natural killer cell mediated cytotoxicity, systemic lupus erythematosus and other pathways (Fig 3E). These results suggested that FERMT2 gene has a widely influence on the global transcriptome.

FERMT2 expression correlated strongly to the MSRB3 expression (r = 0.965, *P* = 2.40E-220), CALD1 (r = 0.964, *P* = 4.21E-219), and ZEB1 (r = 0.939, *P* = 4.53E-177). Significantly, the top 50 positively related genes displayed the high possibility of being important risk genes in CRC.

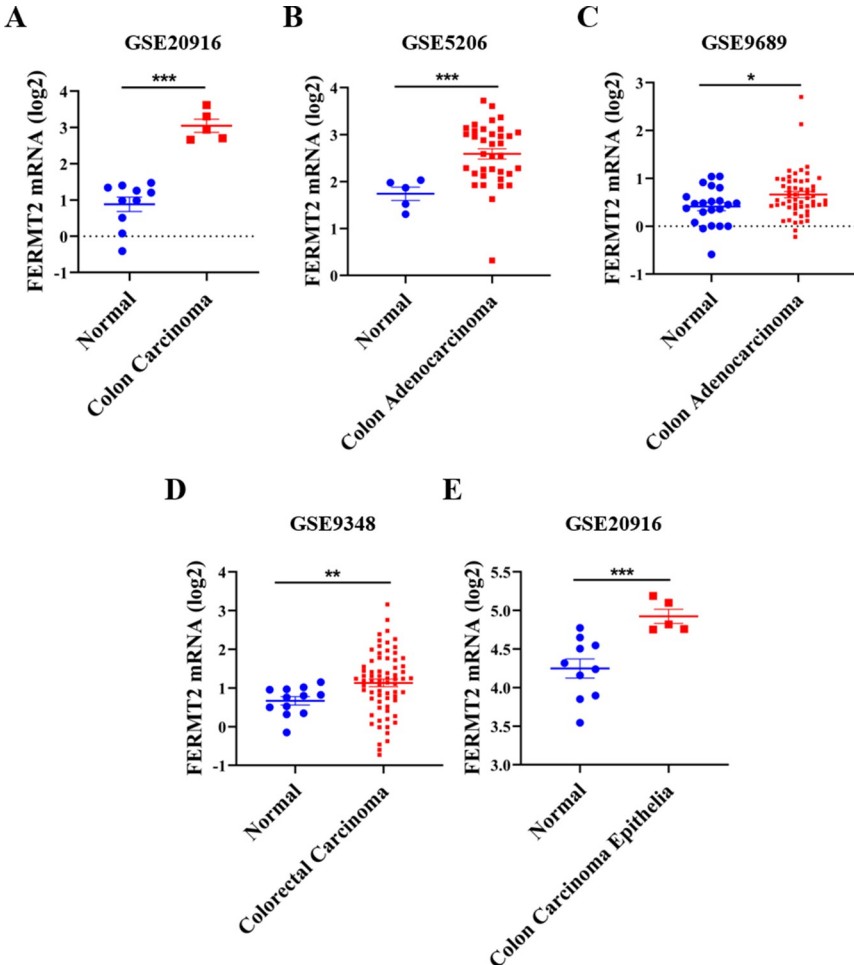

**Fig 1. FERMT2 is highly expressed in the tissues of colorectal cancer (CRC).** (A-E) Comparison of FERMT2 mRNA expression levels between normal and tumor tissues obtained from the Oncomine web tool (Wilcoxon test). $P < 0.05$ was considered to be statistically significant.

Among them, 15/50 genes had a higher hazard ratio ($P < 0.05$) (Fig 3F). Oppositely, 5/50 genes displayed low hazard ratio ($P < 0.05$) in the top 50 negatively associated genes (Fig 3G).

For further exploring the regulators of FERMT2 in CRC, the enrichment of kinases, miR-NAs and transcription factors (TF) related to FERMT2 co-expressed genes were analyzed. The significantly related kinases were PIK3CB and NEK2 (Table 2). The enrichment of miRNA targets was involved in miR-369-3P, miR-448, miR-381, miR-452 and miR-498 (Table 2). Transcription factor enrichment results were shown in Table 2. These data indicated that FERMT2 is involved in the complicated regulating networks in CRC.

**Table 1. Details of the Gene Expression Omnibus (GEO) series included in this analysis.**

| GEO series | Contributor(s) | Tumor | Nontumor | Platform |
|---|---|---|---|---|
| GSE20916 | Skrzypczak M et al, 2010 | 5 | 10 | Affymetrix Human Genome U133 Plus 2.0 Array |
| GSE5206 | Aronow BJ, 2019 | 36 | 5 | Affymetrix Human Genome U133 Plus 2.0 Array |
| GSE9689 | Cardoso J et al, 2008 | 22 | 56 | NKI/CMF Human 18k cDNA spotted microarray |
| GSE9348 | Hong Y | 70 | 12 | Affymetrix Human Genome U133 Plus 2.0 Array |
| GSE20916 | Skrzypczak M et al, 2010 | 5 | 10 | Affymetrix Human Genome U133 Plus 2.0 Array |

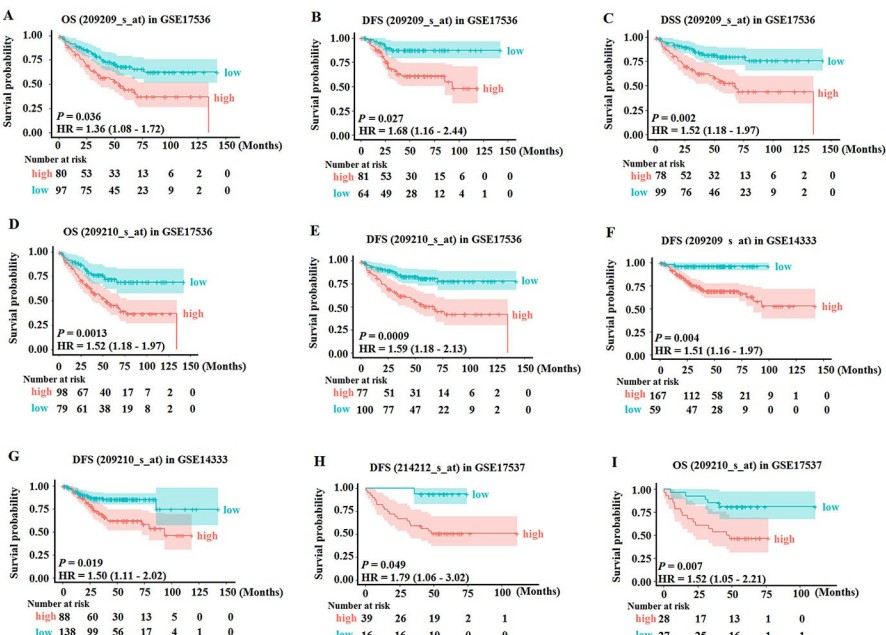

**Fig 2. Up-regulation of FERMT2 is associated with survival outcome in patients with CRC.** (A-C) OS, DFS and DSS in GSE17536(209209_s_at) cohort. (D, E) OS and DFS in GSE17536(209210_s_at) cohort. (F) DFS in GSE14333 (209209_s_at) cohort. (G) DFS in GSE14333(209210_s_at) cohort. (H) OS in GSE17537(214212_s_at) cohort. (I) OS in GSE17537(209210_s_at) cohort. Survival differences are compared between patients with high and low (grouped according to median) expression of FERMT2; the numbers below the figures denote the number of patients at risk in each group; the title of each graphic refers to the project name in PrognoScan web tool; OS: overall survival; DFS: disease-free survival; DSS: disease-specific survival; $P < 0.05$ was considered to be statistically significant.

## Multiple genomic alterations in FERMT2 gene

The somatic mutation profiles of 399 patients with CRC were downloaded from TCGA database. The waterfall plot visualized the detailed information of mutations in each patient (S1A Fig). In the light of further comparison, the majority mutations of different classification categories were SNPs (single-nucleotide polymorphisms), missense variations, and C > T (S1B–S1D Fig). The median of variations in the subjects was 96, and the high limit was 7278 when counting each sample separately (S1E Fig). In the box plots, the numbers of each mutation classification in the samples are displayed (S1F Fig). In addition, the distribution of top 10 mutated genes in CRC are shown in S1G Fig.

Subsequently, we determined the types and frequency of FERMT2 alterations in CRC using the cBioPortal tool. As shown in Table 3, alterations of gene SYNE1, FAT4 and MUC16 in CRC significantly co-occurred with the alterations of gene FERMT2 in AD. These alterations include truncating mutation, missense mutation, amplification, deep deletion, and mRNA upregulation (S2A and S2B Fig). In addition, FERMT2 mRNA expression was significantly positively correlated with SYNE1, FAT4 and MUC16 in CRC (S2C Fig).

## FERMT2 is correlated with the tumor purity and the immune cell infiltration in CRC

To understand if FERMT2 is involved in modulating the tumor immune microenvironment in CRC, the association of FERMT2 expression with the profiles of immune cell infiltration were investigated in CRC using TIMER database. The results indicated that FERMT2 expression had significant negative correlations with tumor purity (Rho = -0.37, $P$ = 1.10e-14) and

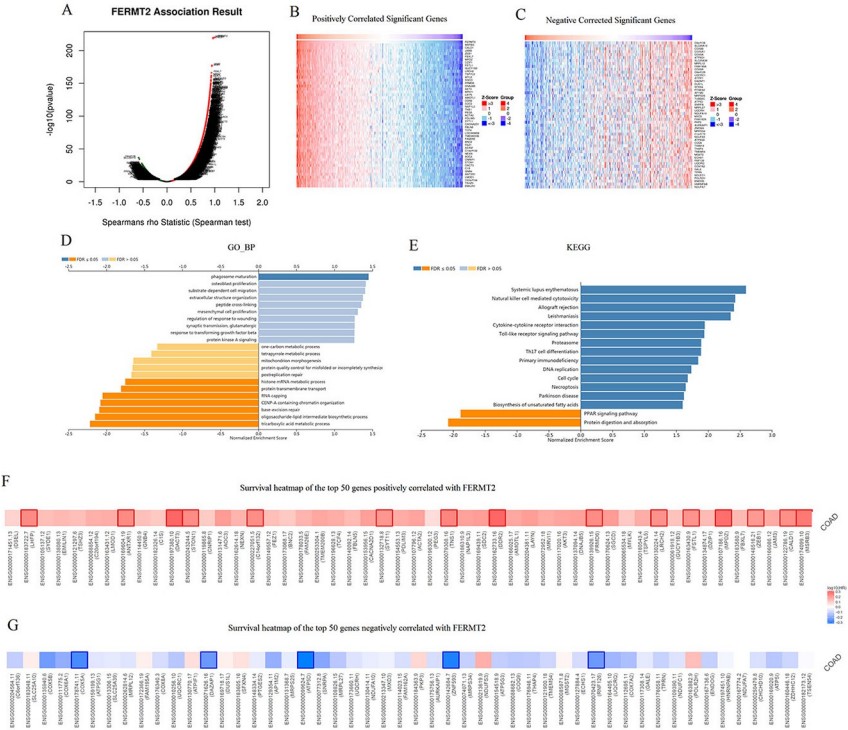

**Fig 3. FERMT2 gene co-expression networks in CRC by LinkedOmics.** (A) The global highly correlated genes with FERMT2 identified by the Spearman test in CRC. Red and green dots represent positively and negatively correlated genes with FERMT2, respectively. (B, C) Heatmaps showing the top 50 genes positively (B) and negatively (C) correlated with FERMT2 in CRC. (D, E) Significantly enriched GO terms (D), the biological process annotations and KEGG pathways (E) of FERMT2 in CRC. (F-G) Survival heatmaps of the top 50 genes positively (F) or negatively (G) correlated with FERMT2 in CRC. The survival heatmaps show the hazard ratios in the logarithmic scale (log10) for different genes. The red and blue blocks denote higher and lower risks, respectively. The rectangles with frames mean the significant unfavorable and favorable results in prognostic analyses ($P < 0.05$). FDR: false discovery rate; KEGG: Kyoto Encyclopedia of Genes and Genomes; GO: Gene Ontology.

the infiltration levels of dominant immune cells, including B cells (Rho = -0.338, $P$ = 8.76e-09), CD4$^+$ T cells (Rho = -0.468, $P$ = 2.44e-16), CD8$^+$ T cells (Rho = -0.214, $P$ = 3.45e-04), NK cells

**Table 2. The kinases, miRNAs and transcription factors-target networks of FERMT2 in CRC.**

| Enriched Category | Geneset | Leading Edge Number | NES* | FDR* |
|---|---|---|---|---|
| Kinase Target | Kinase_PIK3CB | 1 | 1.4592 | 0.08513 |
| | Kinase_NEK2 | 5 | 1.9076 | 0.02608 |
| miRNA Target | GTATTAT, MIR-369-3P | 121 | 1.7575 | 0.015288 |
| | ATATGCA, MIR-448 | 114 | 1.6193 | 0.019237 |
| | CTTGTAT, MIR-381 | 86 | 1.4925 | 0.022374 |
| | TGCAAAC, MIR-452 | 54 | 1.6938 | 0.018005 |
| | GCTTGAA, MIR-498 | 47 | 1.6444 | 0.014851 |
| Transcription Factor | GATAAGR_V$GATA_C | 84 | 1.191 | 0.03554 |
| | V$HNF1_C | 61 | 1.1911 | 0.035694 |
| | V$TBP_01 | 72 | 1.1906 | 0.03573 |
| | V$CRX_Q4 | 73 | 1.1918 | 0.035799 |

*NES: normalized enrichment score; FDR: false discovery rate.

**Table 3. Frequency of co-occurring genomic alterations associated with FERMT2 in patients with CRC.**

| A | B | Neither | A Not B | B Not A | Both | Log2 Odds Ratio | p-Value | q-Value | Tendency |
|---|---|---|---|---|---|---|---|---|---|
| FERMT2 | SYNE1 | 361 | 7 | 139 | 23 | >3 | <0.001 | <0.001 | Co-occurrence |
| | FAT4 | 379 | 12 | 121 | 18 | 2.232 | <0.001 | <0.001 | Co-occurrence |
| | MUC16 | 362 | 14 | 138 | 16 | 1.584 | 0.004 | 0.008 | Co-occurrence |
| | TP53 | 183 | 8 | 317 | 22 | 0.667 | 0.184 | 0.286 | Co-occurrence |
| | TTN | 245 | 12 | 255 | 18 | 0.527 | 0.221 | 0.326 | Co-occurrence |
| | APC | 121 | 9 | 379 | 21 | -0.425 | 0.301 | 0.391 | Mutual exclusivity |
| | KRAS | 275 | 17 | 225 | 13 | -0.098 | 0.507 | 0.508 | Mutual exclusivity |

(Rho = -0.133, *P* = 2.71e-02) and macrophages M2 (Rho = -0.292, *P* = 1.95e-09) in CRC (Fig 4A). However, the above-mentioned immune cell infiltration showed significantly higher enrichment levels in FERMT2-mutated CRC than that in FERMT2-wildtype CRC (Fig 4B). These results suggested that FERMT2 play a role in modulating the tumor immune microenvironment, and is associated with the decrease of anti-tumor immune response in CRC.

## FERMT2 regulates the proliferation and growth of CRC cells

To explore the roles of FERMT2 in the biological behaviors of cells in CRC, the expression of FERMT2 was determined in normal colon epithelial cells NCM460 and five human CRC cell lines, including SW480, SW1116, SW1463, HT29 and LoVo. Our results revealed that the mRNA and protein expression levels of FERMT2 were highly expressed in SW1116 and SW1463 cells compared to NCM460 cells (Fig 5A and 5B). Then, SW1116 and SW1463 cells were used for subsequent experiments. These cells were transfected with negative control or shRNA-FERMT2 plasmid and the knockdown efficiencies of FERMT2 were confirmed by RT-qPCR and western blot

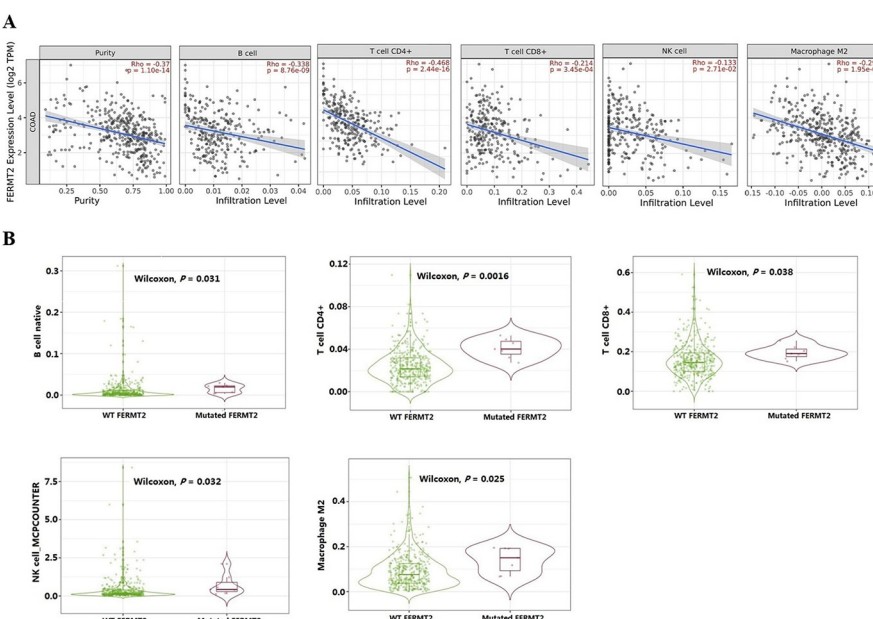

**Fig 4. Correlations of FERMT2 expression with tumor purity and immune cell infiltration levels in CRC.** (A) Correlations of FERMT2 expression with five types of immune cell infiltration obtained from TIMER (purity-corrected Spearman test). (B) Diverse immune signatures showed significantly higher enrichment levels in FERMT2-mutated CRC than that in FERMT2-wildtype CRC (Wilcoxon test, *P* < 0.05).

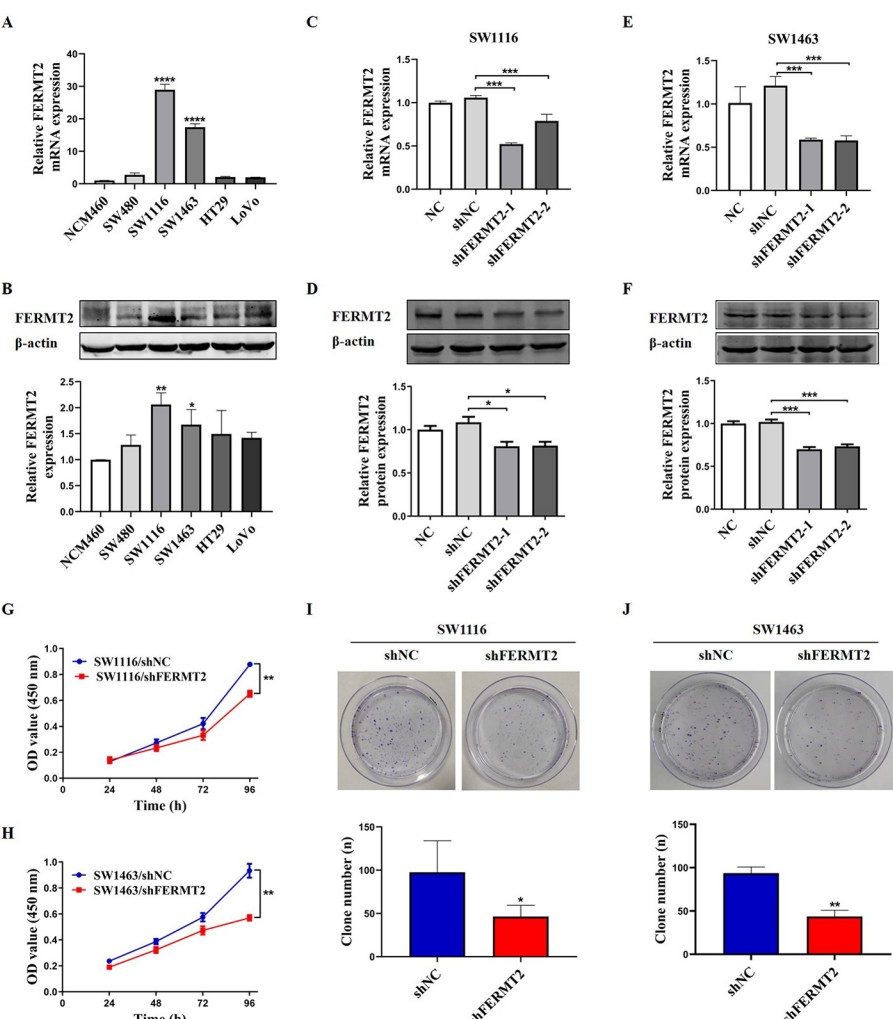

**Fig 5. Knockdown of FERMT2 inhibited cell proliferation and colony formation in CRC cells.** (A, B) The FERMT2 mRNA (A) and protein (B) expression levels in five CRC cell lines and normal colon epithelial cells were detected by RT-qPCR and western blot, respectively. (C-F) The silencing effects on FERMT2 by shRNA in SW1116 (C and D) and SW1463 (E and F) cells were evaluated by RT-qPCR and western blot assay, respectively. (G-H) Effects of FERMT2 silencing on cell proliferation viability in SW1116 and SW1463 cells were measured using a CCK-8 cell-counting kit. (I, J) The colony formation potential of SW1116 and SW1463 cells after FERMT2 knockdown were analyzed with a colony formation assay. All the tests were repeated at least three times. $^{*}P < 0.05$, $^{**}P < 0.01$, $^{***}P < 0.001$ *vs* shNC.

(Fig 5C–5F). Firstly, we examined the expression of β1 integrins in SW1116 and SW1463 cells after knockdown of FERMT2. The results indicated that downregulated FERMT2 reduced the level of β1 integrins in CRC cell lines (S3A–S3D Fig). The cell proliferation assay and colony formation assay showed that silencing FERMT2 remarkably inhibited the cell growth (Fig 5G–5H) and decreased the number of colonies (Fig 5I and 5J) of SW1116 and SW1463 cells. These results suggested that FERMT2 regulates the cell proliferation viability and colony formation ability, and is potentially involved in the tumor oncogenesis and progression of CRC.

## FERMT2 promotes the cell migration, invasion and EMT in CRC

To investigate whether FERMT2 affect the metastatic abilities of CRC cells, the assays of wound healing and transwell were conducted in the SW1116 and SW1463 cells knocked

down of FERMT2. Our findings revealed that knockdown of FERMT2 significantly inhibited the migration and invasion ability of SW1116 and SW1463 cells (Fig 6A–6D). The expression of EMT biomarkers was determined and the results indicated that knockdown of FERMT2 evidently increased the levels of E-cadherin protein (a marker of epithelial) and reduced the protein levels of N-cadherin and Vimentin (markers of mesenchymal) in SW1116 cells (Fig 6E and 6F), as well as in SW1463 cells (Fig 6G and 6H). These results suggested that FERMT2 promotes the cell migration, invasion and EMT in the biological process of CRC.

## FERMT2 promotes the migration and invasion of cells in CRC via activating Wnt/β-catenin pathway

Several studies have revealed that the pathway of Wnt/β-catenin was involved in the process of cell proliferation, migration, and invasion in CRC [24, 35]. In this study, the analysis of Spearman's correlation was conducted and the results showed that the expression of FERMT2 was

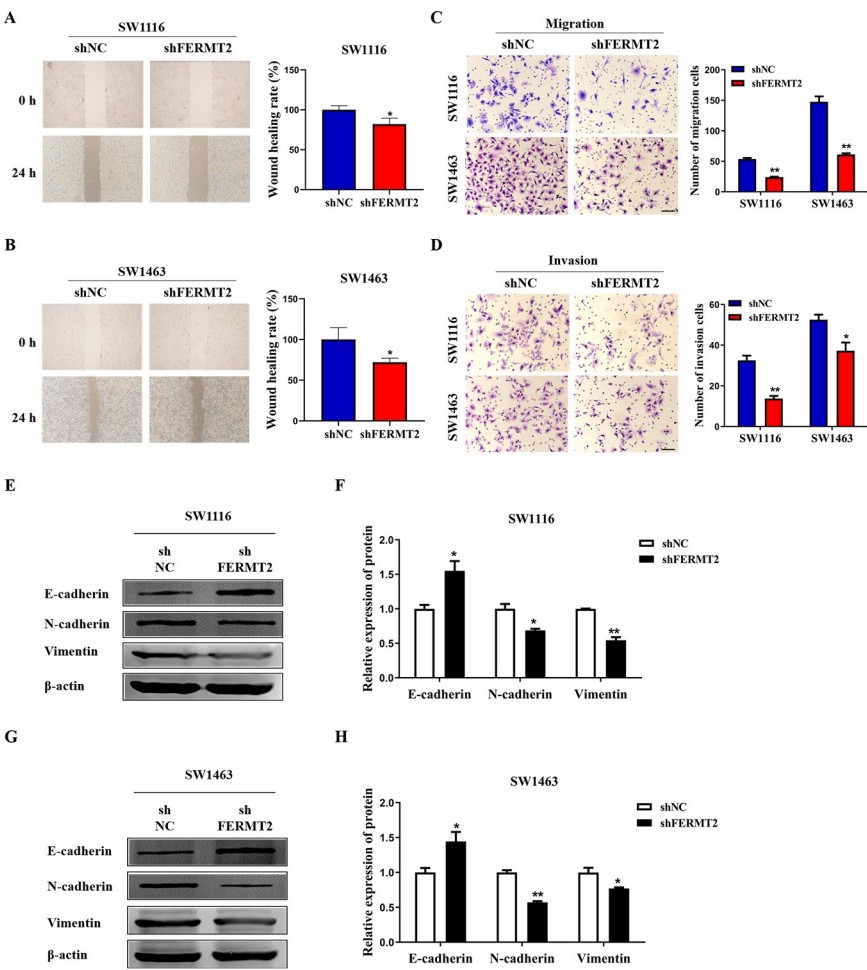

**Fig 6. Knockdown of FERMT2 inhibited the migration, invasion and EMT in SW1116 or SW1463 cells.** (A-D) The migration ability of SW1116 or SW1463 cells after FERMT2 knockdown were detected by wound healing assay (A, B) and transwell assay (C, D). Scale bars: 100 μm. (E-H) The expression of E-cadherin, N-cadherin and Vimentin in SW1116 (E-F) or SW1463 (G, H) cells after FERMT2 knockdown were measured by western blot assay. All the tests were repeated at least three times. $^{*}P < 0.05$, $^{**}P < 0.01$ vs shNC.

positively correlated with CTNNB1 (β-catenin) in the TCGA-COAD database (Fig 7A). The results by western blot suggested that β-catenin and cyclin D1 (a target protein of Wnt/β-catenin signaling) levels were decreased following FERMT2 knockdown in SW1116 (Fig 7B) and SW1463 (Fig 7C) cells. These results demonstrated that knockdown of FERMT2 inhibited Wnt/β-catenin signaling in CRC cells. To validate if β-catenin activation plays a critical role in

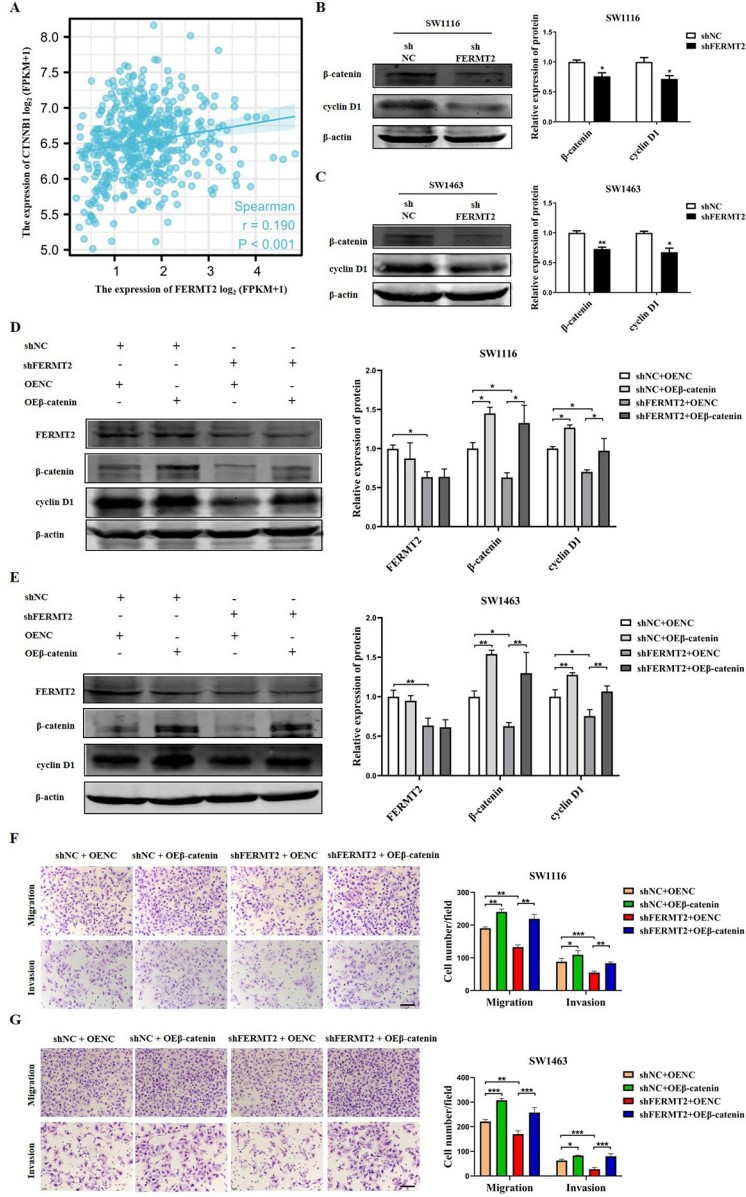

**Fig 7. FERMT2 promoted migration and invasion of CRC cells through activating Wnt/β-catenin signaling pathway.** (A) Spearman's correlation analysis between FERMT2 and CTNNB1 (β-catenin) expression in patients with CRC from TCGA-COAD dataset. (B, C) The expression levels of β-catenin and cyclinD1 protein in SW1116 (B) and SW1463 (C) cells after FERMT2 knockdown were detected by Western blot. (D, E) The expression of FERMT2, β-catenin and cyclinD1 protein in SW1116 (D) and SW1463 (E) cells were measured by western blotting after co-transfection of shFERMT2 and β-catenin plasmids. (F, G) Overexpression of β-catenin significantly reversed the influence of FERMT2 knockdown on the cell migration and invasion in SW1116 (F) and SW1463 (G) cells as revealed by the transwell assay. Scale bars: 100 μm. All the tests were repeated at least three times. $^{*}P < 0.05$, $^{**}P < 0.01$, $^{***}P < 0.001$ *vs* shNC.

the process of FERMT2 promoting the migration and invasion of CRC cells, we co-transfected the CRC cells with the FERMT2 knockdown and the β-catenin overexpression plasmid. Our results showed that β-catenin upregulation resulted in an increase of cyclin D1 expression, but did not affect FERMT2 levels (Fig 7D and 7E), and the inhibitory effects on the cell migration or invasion induced by knockdown of FERMT2 were rescued by overexpression of β-catenin (Fig 7F and 7G) in SW1116 and SW1463 cells. These results suggested that FERMT2 promotes the cell migration and invasion through activation of Wnt/β-catenin pathway in CRC.

## AD model cells inhibit the migration, invasion, EMT and Wnt/β-catenin signaling of CRC cells

To further investigate the interactive effects between AD model cells and CRC cells, the migration, invasion, EMT and the proteins related to Wnt/β-catenin signaling pathway were determined in CT26 cells co-cultured with $A\beta_{1-42}$ treated HT22 cells. Firstly, the AD cell model was developed by incubating HT22 cells with different concentrations of $A\beta_{1-42}$ for 24 h. The cell viability of HT22 cells were reduced in a concentration dependent manner and a thirty percent reduction of the control cell viability was observed in the cells treated with 40 μM $A\beta_{1-42}$ (Fig 8A). Therefore, the concentration of 40 μM $A\beta_{1-42}$ was used to develop AD cell model in the subsequent experiments. Then, the FERMT2 expression was tested by western blot assay and the result displayed that the level of FERMT2 protein was reduced in the AD model cells (Fig 8B and 8C). Furtherly, the migration and invasion of CT26 cells were inhibited by co-cultured HT22 cells treated with $A\beta_{1-42}$ (Fig 8D and 8E).

To explore the possible mechanisms by which the migration and invasion abilities of CT26 cells were inhibited, western blot assay was conducted to determine the biomarkers of EMT and the proteins related to Wnt/β-catenin signaling pathway in CT26 cells after co-cultured with the AD model cells. The results revealed that the protein level of E-cadherin was increased, but N-cadherin and Vimentin were decreased in CT26 cells (Fig 8F and 8G). Meanwhile, the protein levels of FERMT2, β-catenin and cyclin D1 were downregulated in CT26 cells (Fig 8H and 8I). These results suggested that AD model cells inhibited the migration, invasion and EMT of CRC cells possibly through suppressing the Wnt/β-catenin signaling pathway.

## Downregulation of FERMT2 in the AD model cells contributes to the inhibited migration and invasion of CRC cells

To validate whether FERMT2 in AD model cells plays a role in inhibiting the migration and invasion of CT26, HT22 cells were transfected with the FERMT2 overexpression plasmid before the $A\beta_{1-42}$ (40 μM) exposure for 24 h and the protein levels of FERMT2 were detected by western blot assay (Fig 9A). We found that the level of FERMT2 was decreased in AD model cells overexpressing FERMT2 compared to the control cells (Fig 9B). In addition, the cell co-culture results showed that the inhibited effects on the migration and invasion of CT26 cells by HT22 cells treated with $A\beta_{1-42}$ were reversed when the AD model cells overexpressing FERMT2 (Fig 9C and 9D). These results suggest that downregulation of FERMT2 in AD model cells plays a part in inhibiting the migration and invasion of CRC cells.

## AD model cells inhibit the migration and invasion of CRC cells through downregulating Wnt/β-catenin signaling pathway

To explore the mechanisms underlying the AD model cells-inhibited migration and invasion abilities of CRC cells, we transfected CT26 cells with β-catenin overexpression plasmid 24 h

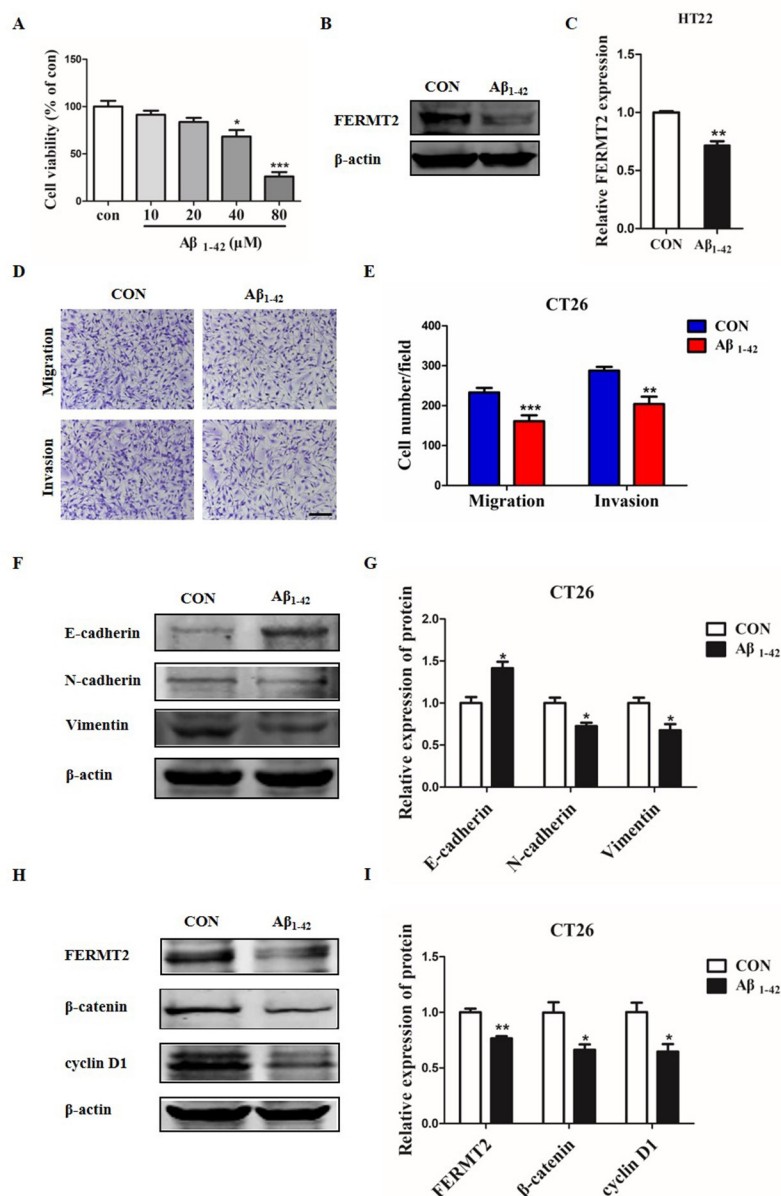

**Fig 8. Aβ$_{1-42}$ treated HT22 cells inhibited the migration, invasion, EMT and the proteins related to Wnt/β-catenin signaling pathway in CT26 cells.** (A) CCK-8 assay was used to measure the cell viability of HT22 cells after treatment with different concentrations of Aβ$_{1-42}$ for 24 h. (B, C) The expression of FERMT2 in the 40 μM Aβ$_{1-42}$-treated cells was measured by western blotting. (D, E) The migration and invasion ability of CT26 cells co-cultured with the Aβ$_{1-42}$-treated cells were determined by transwell assays. Scale bars: 100 μm. (F, G) The expression of E-cadherin, N-cadherin and Vimentin in CT26 cells co-cultured with the Aβ$_{1-42}$-treated cells were detected by western blotting. (H, I) The expression of FERMT2, β-catenin and cyclin D1 protein in CT26 cells co-cultured with the Aβ$_{1-42}$-treated cells were detected by western blotting. $^{*}P < 0.05$, $^{**}P < 0.01$, $^{***}P < 0.001$ *vs* CON.

before co-cultured with the AD model cells. The protein level of β-catenin was analyzed by western blot (Fig 10A and 10B). Afterwards, we co-cultured the Aβ$_{1-42}$ treated HT22 cells with the CT26 cells overexpressing β-catenin. The results showed that the inhibition of migration and invasion abilities of CT26 cells by the Aβ$_{1-42}$ treated HT22 cells were reversed by overexpressing β-catenin in CT26 cells (Fig 10C and 10D). Moreover, the AD model cells-induced

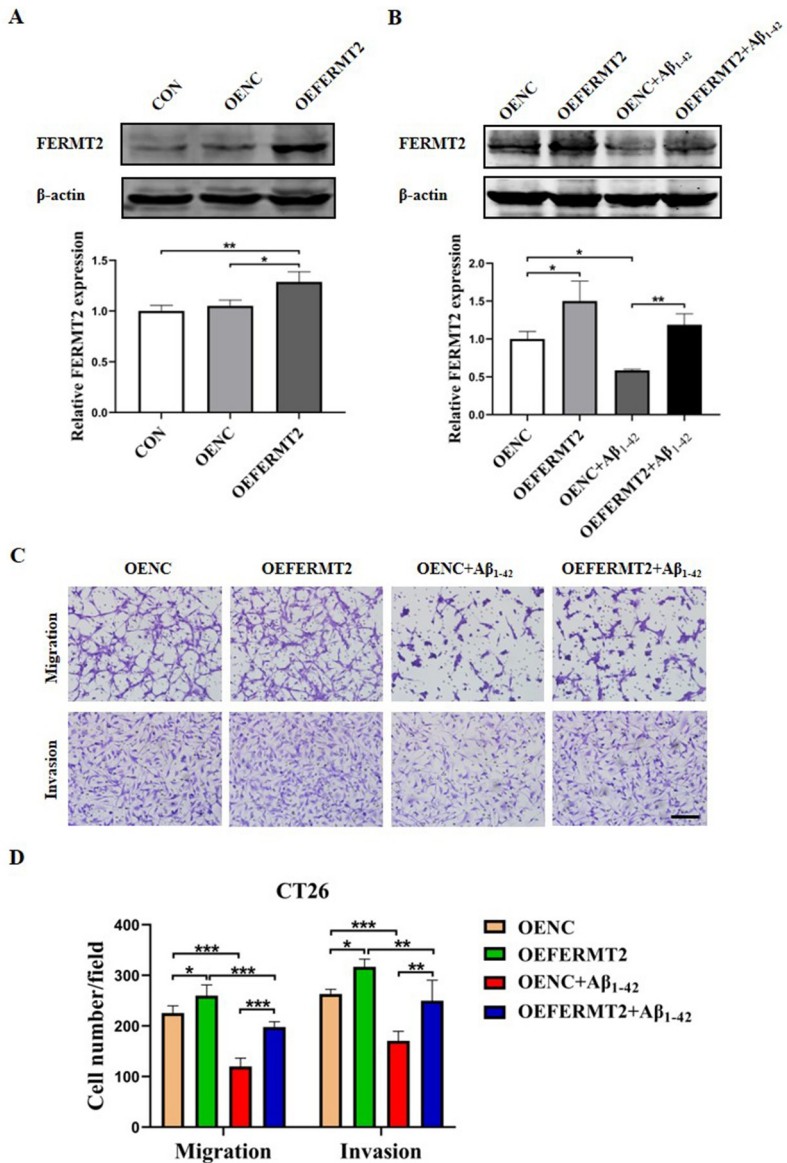

**Fig 9. FERMT2 in Aβ$_{1-42}$ treated HT22 cells participated in the regulation of migration and invasion in CT26 cells.** (A, B) The HT22 cells were treated with or without Aβ$_{1-42}$ (40 μM) and transfected with FERMT2 overexpression or control plasmid. The proteins of FERMT2 and β-actin were detected by western blot analysis. (C, D) Overexpression of FERMT2 attenuated the Aβ-induced inhibition on the migration and invasion of CT26 cells. Scale bars: 100 μm. $^*P < 0.05$, $^{**}P < 0.01$, $^{***}P < 0.001$, $^{****}P < 0.0001$.

reduction of β-catenin and cyclin D1 proteins were abolished by overexpressing β-catenin in CT26 cells (Fig 10E). These results indicate that the interactive effects of AD model cells on the migration and invasion of CRC cells are dependent on the Wnt/β-catenin signaling pathway.

## Discussion

More and more attention has been paid to the research of disease-disease relationships, which can help us to better understand the true nature of diseases, and ultimately to improve the outcomes of the patients. Jörg Menche et al had mapped the disease-disease relationships through

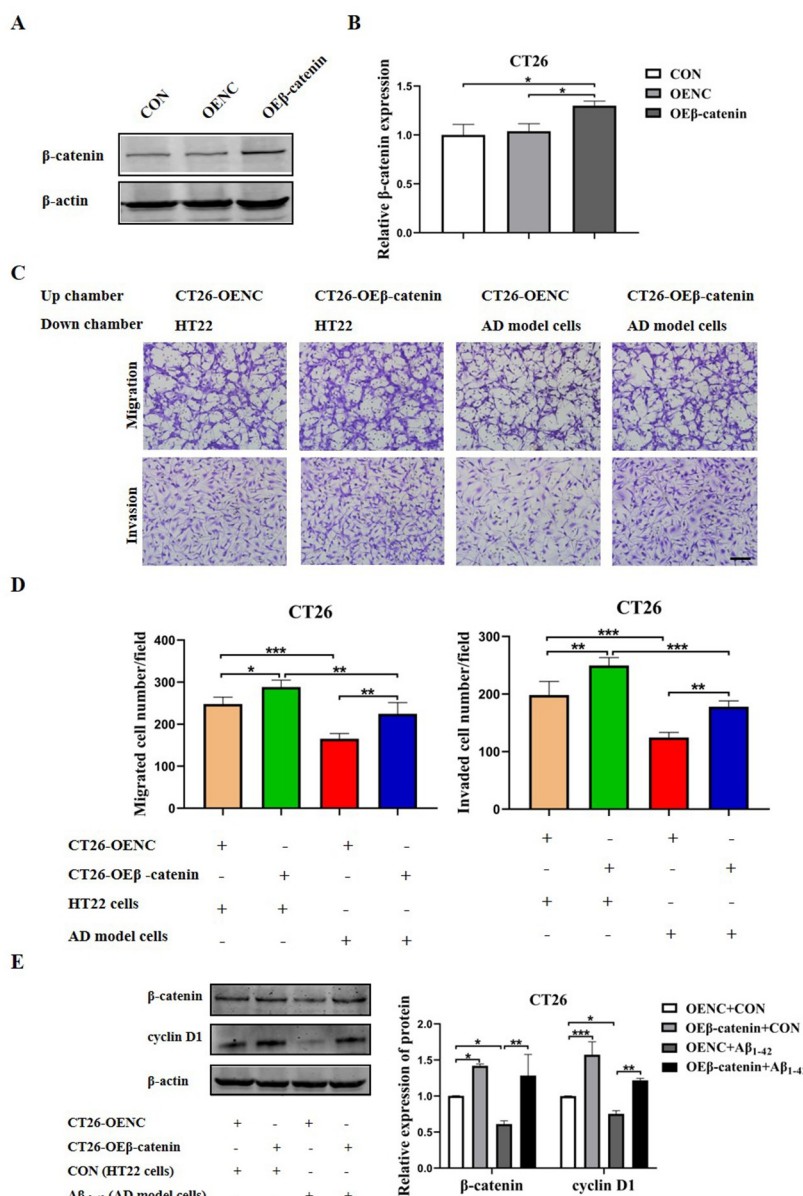

**Fig 10. Aβ₁₋₄₂ treated HT22 cells inhibited the migration and invasion of CT26 cells through the Wnt/β-catenin signaling pathway.** (A, B) CT26 cells were transfected with β-catenin overexpression plasmid. The protein levels of β-catenin and β-actin were detected by Western blot analysis. (C, D) The HT22 cells were treated with or without Aβ₁₋₄₂ (40 μM) and co-cultured with CT26 cells transfected with β-catenin overexpression or control plasmid. The migration and invasion abilities of CT26 cells were determined by the transwell assays. Scale bars: 100 μm. (E) The CT26 cells were transfected with β-catenin overexpression or control plasmid and co-cultured with or without the AD model cells. The levels of β-catenin and cyclin D1 proteins in CT26 cells were measured by western blot assay. $^{*}P < 0.05$, $^{**}P < 0.01$, $^{***}P < 0.001$.

a platform of interactome to predict the molecular commonalities among diseases related to phenotype, which is of great help for exploring the molecular mechanisms of human diseases [36]. AD and cancer are both associated with aging, but can be regarded as opposite phenomena. Many aspects are upregulated to sustain the cell growth and survival in cancers, but are downregulated contributing to neurodegeneration in AD [37]. Unlike the dysregulation growth control and genetic mutations related to cancers, AD is characterized by damage and

death of brain neurons without genomic changes [1]. Although multiple biological pathways controlling oncogenesis, such as oxidative stress, inflammatory response and metabolic dysfunction, have been associated with AD, the underlying molecular mechanisms are not well elucidated. Understanding the relationship between AD and cancers will not only help us to find new strategies for prevention and treatment of both diseases, but also impel us to re-understand the human disease and health with a holistic concept.

FERMT2, a genetic risk factor for AD, has a link with brain amyloidosis in stage-dependent manner, which is most remarkable in the stage of mild cognitive impairment [38]. Recent study has reported that FERMT2 expression is related to the modulation of Aβ accumulation and tau hyperphosphorylation in human neurons [39]. However, FERMT2 is upregulated in many kinds of cancers and modulates cancer progression through various molecular pathway. For example, FERMT2 promotes the invasion, metastasis and EMT of HCC via Wnt/β-catenin pathway [24]. In breast cancer, FERMT2 regulates tumor growth through promoting CSF-1-mediated infiltration of macrophages [40]. However, the roles and underlying mechanism of AD-associated gene FERMT2 in the progression of CRC remain unknown.

Tumorigenesis of CRC is generally associated with genetic alterations, life-style and dietary risk factors [41]. To explore the roles of FERMT2 in CRC formation, firstly the Oncomine analysis results demonstrated that FERMT2 mRNA expression was upregulated in tumor tissues from various types of CRC patients compared with normal colon tissues. Although the results of FERMT2 expression in colorectal cancers are inconsistent from different databases, and the reasons may be related to pathological classification, progress, patient population, detection window and other factors. In this study, we firstly analyzed the expression of FERMT2 protein using the public database, and then we have verified the results in patients with colorectal cancer from our cohort, as well as in colorectal cancer cells. Both the bioinformatics analysis results and our detection indicated that the FERMT2 expression was up-regulated in CRC tissues and CRC cells. Moreover, knockdown of FERMT2 significantly suppressed the proliferation and clone formation of CRC cells. In addition, high expression of FERMT2 was remarkably associated with OS, DFS and DSS in many cohorts. The FERMT2 co-expression network implicated that the functional consequence of FERMT2 mainly include cytotoxicity mediated by natural killer cell, interaction between cytokine-cytokine receptors, Toll-like receptor pathway, while it inhibits the PPAR signal pathway, protein digestion and absorption, and RNA capping process. These findings are identical with the signaling pathways implicated in the tumorigenesis of CRC [42, 43]. Thus, our results suggested that FERMT2 expression has a link with the occurrence of CRC and may act as a potential prognostic indicator in CRC.

Abundant evidence indicates that genomic alteration occurs frequently in human neoplasms and the mutation of specific genes are associated with the immunotherapy response, drug treatment and a poor prognosis [44]. Our study revealed that alterations in the CRC genes SYNE1, FAT4 and MUC16 were co-occurred significantly with alterations in the AD risk gene FERMT2. These data demonstrate that genes of co-occurrent alterations play critical roles in the development of CRC. The microenvironment of tumor consists of the non-cancerous cells, including immune cells, which exist in and around a tumor. It may play an important part in tumor biology and have a strong impact on the genomic analysis of tumor samples [45]. More notably, co-occurrence genes are related to genetic diversity and tumor microenvironment processes [46]. These results demonstrate that FERMT2 participates in the modulating the tumor immune microenvironment, and it is associated with the decrease of anti-tumor immune response in CRC.

Numerous evidence has suggested that the main cause of death is metastasis in several malignant tumors, and the distant metastasis is correlated with the poor survival of patients

with CRC [47]. Integrins, composed of α- and β-subunits, mediate recognition and adhesion between cells or between cells and extracellular matrix. Several studies have shown that β1 integrins can regulate the proliferation, migration and invasion of tumor cells through different signal pathways [48, 49]. FERMT2 has been shown to modulate β1 integrins activity in cancers. Here we examined the expression of β1 integrins on SW1116 and SW1463 cells after knockdown of FERMT2 by FACS. The results indicated that knockdown of FERMT2 reduced the expression of β1 integrin in CRC cell lines, suggesting that β1 integrin subunit is involved in the FERMT2 mediated progression in CRC. Given that FERMT2 is upregulated in CRC tissues and cells, we found that knock down of FERMT2 inhibited the migration and invasion capacity of CRC cells significantly. EMT represents a process in which the epithelial cells lose apical-basal polarity and participate in the migration and invasion in various cancer cells, including CRC [50]. Our study indicated that silencing FERMT2 inhibited CRC cell invasion and migration by blocking the process of EMT. These results illustrate the role of FERMT2 in promoting the migration and invasion ability of CRC cells.

Abnormally activated Wnt/β-catenin pathway is ubiquitous in a variety of tumors, and upregulated Wnt/β-catenin pathway promotes tumor progression and deterioration [51]. It is reported that FERMT2 participated in the cancer progression and metastasis through regulating numerous signaling pathways, which are essential for survival, proliferation, migration and invasion of tumor cells [52]. However, little knowledge has been known about the mechanism by which FERMT2 promotes CRC migration and invasion until now. Therefore, we firstly analyzed the relationship between FERMT2 and β-catenin in TCGA-COAD dataset and the results indicated that FERMT2 is positively correlated with the expression of β-catenin. In addition, the expression of β-catenin and its target protein cyclin D1 were decreased following knockdown of FERMT2 in CRC cells. These results indicated that FERMT2 activated Wnt/β-catenin pathway in CRC. Furthermore, overexpressing of β-catenin was able to rescuse the effect of FERMT2 on CRC cells. As reported that Wnt/β-catenin signaling pathway is one of the common upstream signals of EMT, and activation of this signaling pathway often induces EMT of cells in cancer [53, 54]. FERMT2 may induce the expression of cyclin D1 and EMT related proteins by regulating the Wnt /β-catenin signaling pathway and promote the migration and invasion of CRC cells. It is indicated that β-catenin is essential for FERMT2 promoting the migration and invasion in CRC cells. Meanwhile, Wnt/β-catenin signaling is required for FERMT2-induced CRC progression.

The tumor microenvironment plays critical roles in tumor initiation, progression and metastasis. Specially, nerve fibers in the microenvironment around tumor regulate the tumorigenesis and dissemination. Philippe Mauffrey found that neural progenitors from the central nervous system drove neurogenesis and promoted tumor development and metastasis in prostate cancer, suggesting that nerve fibers in the tumor microenvironment are significantly associated with tumor initiation and progression [55]. More importantly, neurogenesis in CRC appeared to play a critical role in CRC progression and function as an independent predictor of poor outcomes [56]. To understand the interaction between AD model cells and CRC cells, we co-cultured Aβ$_{1-42}$ treated HT22 cells with CT26 cells in the current study, and we found that Aβ$_{1-42}$ treated HT22 cells induced downregulation of FERMT2 and inhibited the migration, invasion and EMT in co-cultured CT26 cells. Wnt signaling can regulate the axonal regeneration and is activated in various cancers, consequently, the process of neurogenesis and tumorigenesis may share the Wnt signaling pathway. In the current study, we found that AD model cell inhibited Wnt pathway of CT26 cells, whereas the FERMT2-induced reduction of β-catenin and cyclin D1 protein was recovered by overexpressing β-catenin in CT26 cells. Taken together, these results strongly suggest that FERMT2 in AD model cells plays a role in inhibiting the migration and invasion of CRC cells via Wnt/β-catenin signaling pathway.

## Conclusion

In summary, this study demonstrated that the Alzheimer's genetic risk factor FERMT2 promotes the progression of CRC through Wnt/β-catenin signaling pathway and contributes to the inverse correlation between AD and CRC, which provides experimental foundation for further exploring the negative correlation mechanism between AD and cancers. Our findings not only provide new biological targets for improving diagnosis and treatment of CRC, but also reveal novel mechanisms underlying the negative correlation between AD and cancer pathogenesis.

## Supporting information

**S1 Fig. Analyses of somatic mutation profiles in CRC samples.** (A) Waterfall plot of detailed mutation information of top 20 genes in each sample, with various color annotations to distinguish different mutation types. (B-D) According to different classification categories, missense mutation, SNP, and C > T mutation accounted for the overwhelming majority. (E) The total mutation number in each sample. (F) Box plots of each variant classification in each sample. (G) Top 10 mutated genes in CRC. SNP, single nucleotide polymorphism; SNV, single nucleotide variants.
(TIF)

**S2 Fig. Frequency of co-occurring genetic alterations in FERMT2, SYNE1, FAT4 and MUC16, and the correlation between FERMT2 with SYNE1, FAT4 or MUC16 in CRC patients (cBioPortal).** (A, B) OncoPrint of FERMT2 co-occurring alterations in CRC cohort. The different types of genetic alterations are highlighted in different colors. (C) Correlations between FERMT2 with SYNE1, FAT4 or MUC16 at mRNA expression levels in patients with CRC.
(TIF)

**S3 Fig. Knockdown of FERMT2 inhibited the expression of β1 integrins in SW1116 or SW1463 cells.** (A-D) The expression of β1 integrins in SW1116 cells (A, B) or SW1463 cells (C, D) after FERMT2 knockdown was measured by FACS. $^{*}P < 0.05$ vs shNC.
(TIF)

**S1 Table. The detailed description of the genes co-expressed with FERMT2.**
(XLSX)

**S1 File.**
(DOCX)

**S1 Raw images.**
(PDF)

## Acknowledgments

The authors thank Key Laboratory for Colorectal Cancer Precision Diagnosis and Treatment of Hebei province for providing CRC cells.

## Author Contributions

**Conceptualization:** Shunjiang Xu.

**Data curation:** Zhaoyu Gao, Lei Jiang.

**Formal analysis:** Zhaoyu Gao.

**Funding acquisition:** Shunjiang Xu.

**Methodology:** Lei Jiang.

**Project administration:** Rui Zhang.

**Resources:** Xia Jiang.

**Software:** Wanchang Wang.

**Supervision:** Shunjiang Xu.

**Validation:** Wenzhen Xia, Yushi Qin, Di Zhang, Pei Tian, Wanchang Wang, Qi Zhang.

**Visualization:** Rui Zhang, Nan Zhang.

**Writing – original draft:** Wenzhen Xia.

**Writing – review & editing:** Rui Zhang, Nan Zhang, Shunjiang Xu.

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
