## [Decision Letter · Decision Letter 0]

12 Oct 2022

PONE-D-22-25240Alzheimer’s risk factor FERMT2 promotes the progression of colorectal carcinoma via Wnt/β-catenin signaling pathway and contributes to the negative correlation between Alzheimer and cancerPLOS ONE

Dear Dr.Xu,

Thank you for submitting your manuscript to PLOS ONE. After careful consideration, we feel that it has merit but does not fully meet PLOS ONE’s publication criteria as it currently stands. Therefore, we invite you to submit a revised version of the manuscript that addresses the points raised during the review process.

Dear Doctor Xu

Your manuscript has been evaluated by a reviewer and by me, as Academic Editor. It is a complete manuscript with a lot of data. However, some issues need to be addressed.

(1) The authors should explain why - at least in the discussion part - some websites such as GEPIA (http://gepia.cancer-pku.cn/detail.php?gene=FERMT2) indicate that FERMT2 is under expressed in colorectal cancers compared to healthy tissue. These results seem to be in disagreement with your observations.

(2) The authors should analyze the impact of FERMT2 knockdown on the expression or activation of beta1 integrin subunit.

(3) The authors should respond to the reviewer's comment concerning the purity of the AB1-42 peptide

(4) Collective cell migration plays an important role in the progression of colorectal cancer. However, the results presented in the manuscript only show single cell migration. The impact of FERMT2 in the collective migration could be assessed by the authors using a wound healing test.Dear Doctor Xu

We look forward to receiving your revised manuscript.

Kind regards,

Frédéric André

Academic Editor

PLOS ONE

Additional Editor Comments :

Dear Doctor Xu

Your manuscript has been evaluated by a reviewer and by me, as Academic Editor. It is a complete manuscript with a lot of data. However, some issues need to be addressed.

(1) The authors should explain why - at least in the discussion part - some websites such as GEPIA (http://gepia.cancer-pku.cn/detail.php?gene=FERMT2) indicate that FERMT2 is under expressed in colorectal cancers compared to healthy tissue. These results seem to be in disagreement with your observations.

(2) The authors should analyze the impact of FERMT2 knockdown on the expression or activation of beta1 integrin subunit.

(3) The authors should respond to the reviewer's comment concerning the purity of the AB1-42 peptide

(4) Collective cell migration plays an important role in the progression of colorectal cancer. However, the results presented in the manuscript only show single cell migration. The impact of FERMT2 in the collective migration could be assessed by the authors using a wound healing test.

Reviewers' comments:

Reviewer's Responses to Questions

**Comments to the Author**

1. Is the manuscript technically sound, and do the data support the conclusions?

Reviewer #1: Yes

2. Has the statistical analysis been performed appropriately and rigorously? 

Reviewer #1: Yes

3. Have the authors made all data underlying the findings in their manuscript fully available?

Reviewer #1: Yes

4. Is the manuscript presented in an intelligible fashion and written in standard English?

Reviewer #1: Yes

5. Review Comments to the Author

Reviewer #1: This manuscript examine the relationship FERMT2, colorectal carcinoma (CRC) and Alzheimer disease (AD). FERMT2 levels are elevated in CRC and with poor prognostic outcomes. Knockdown of FERMT2 suppresses key tumorigenic responses in CRC cell lines, and these effects are linked to suppression of Wnt/β-catenin signaling. Treatment with the AD associated peptide, AB1-42 peptide, decreased FERMT2 levels in CRC levels and Wnt signaling, suggesting a basis for an inverse correlation between CRC and AD. Overall, the study is well-done, and an extensive body of data are presented. However, some issues need to be addressed:

1) By far, the most prominent role of FERMT2 is its regulation of integrin activity, much more so than its effects on Wnt/β-catenin signaling. Does knockdown of FERMT2 in CRC cells reduce the expression or activation of integrins on CRC cell lines? It would be sufficient to examine the expression and activity of β1 integrins on the cells by FACS using commercially available antibodies.

2) The AB1-42 is reported to be high purity but was the peptide tested for non-peptide contaminants. Was the peptide free of endotoxin or other non-peptide that could activate/inactivate cells?

6. PLOS authors have the option to publish the peer review history of their article (what does this mean?). If published, this will include your full peer review and any attached files.

Reviewer #1: No

---

## [Author Response · Author response to Decision Letter 0]

19 Nov 2022

Dear reviewers and editors,

Thank you very much for giving us an opportunity to revise our manuscript entitled “Alzheimer’s risk factor FERMT2 promotes the progression of colorectal carcinoma via Wnt/β-catenin signaling pathway and contributes to the negative correlation between Alzheimer and cancer”. (PONE-D-22-25240). We really appreciate the editors and reviewers for your helpful and constructive comments and suggestions on our manuscript.

We have studied the reviewer’s comments carefully and have made revision which marked in red in the paper. We have tried our best to revise our manuscript according to the comments. We would like to express our great appreciation to editors and reviewers for comments on our paper. Below the comments of the reviewers are response point by point and the revisions are indicated. We hope our revised manuscript will meet with the approval of editors to publish it in PLoS ONE. Thank you!

Best regards.

Yours sincerely,

Shunjiang Xu

Response to reviewers and editors

Comment (1): The authors should explain why - at least in the discussion part - some websites such as GEPIA (http://gepia.cancer-pku.cn/detail.php?gene=FERMT2) indicate that FERMT2 is under expressed in colorectal cancers compared to healthy tissue. These results seem to be in disagreement with your observations.

Response: We are grateful to you for this suggestion. Indeed, the results of FERMT2 expression in colorectal cancers are inconsistent from different databases. The reasons may be related to pathological classification, progress, patient population, detection window and other factors. In this study, we firstly analyzed the expression of FERMT2 protein using the public database (Oncomine), and then we have verified the results in patients with colorectal cancer from our cohort, as well as in colorectal cancer cells. Both the bioinformatics analysis results and our detection indicated that the FERMT2 expression was up-regulated in CRC. We have discussed this disagreement in our revised manuscript. 

Comment (2): The authors should analyze the impact of FERMT2 knockdown on the expression or activation of beta1 integrin subunit. (By far, the most prominent role of FERMT2 is its regulation of integrin activity, much more so than its effects on Wnt/β-catenin signaling. Does knockdown of FERMT2 in CRC cells reduce the expression or activation of integrins on CRC cell lines? It would be sufficient to examine the expression and activity of β1 integrins on the cells by FACS using commercially available antibodies.)

Response: Thank you very much for your valuable comments. Integrins, composed of α- and β-subunits, mediate recognition and adhesion between cells or between cells and extracellular matrix. Several studies have shown that β1 integrins can regulate the proliferation, migration and invasion of tumor cells through different signal pathways. FERMT2 has been shown to modulate β1 integrins activity in cancers. According to the reviewer’s suggestion, we have examined the expression of β1 integrins on SW1116 and SW1463 cells after knockdown of FERMT2 by FACS. The results indicated that knockdown of FERMT2 reduced the expression of β1 integrin in CRC cell lines, suggesting that β1 integrin subunit is involved in the FERMT2 mediated progression in CRC. We have provided the results as supplementary data (S3 Fig A-D) and discussed it in our revised manuscript.

S3 Fig. Knockdown of FERMT2 inhibited the expression of β1 integrins in SW1116 or SW1463 cells. (A-D) The expression of β1 integrins in SW1116 cells (A, B) or SW1463 cells (C, D) after FERMT2 knockdown was measured by FACS. *P < 0.05 vs shNC.

Comment (3): The authors should respond to the reviewer's comment concerning the purity of the Aβ1-42 peptide. (The Aβ1-42 is reported to be high purity but was the peptide tested for non-peptide contaminants. Was the peptide free of endotoxin or other non-peptide that could activate/inactivate cells?)

Response: Thank you for your question. Firstly, we didn’t test the non-peptide contaminants of the Aβ1-42 peptide we used in this study. According to the certificate of analysis, the purity of Aβ1-42 is 98.93 %. In addition, we have contacted the biosynthesis company and they said that the Aβ1-42 peptide was synthesized by chemical synthesis method. The materials used for chemical synthesis include amino acids, resins, and some other chemical reagents, but do not include Escherichia coli, which can produce endotoxin. Therefore, the Aβ1-42 peptide is free of endotoxin or other non-peptide that could activate/inactivate cells. 

Comment (4): Collective cell migration plays an important role in the progression of colorectal cancer. However, the results presented in the manuscript only show single cell migration. The impact of FERMT2 in the collective migration could be assessed by the authors using a wound healing test.

Response: Thank you for your constructive comment. According to the suggestion of the reviewers, we have detected the ability of collective migration in SW1116 and SW1463 cells after knockdown of FERMT2 by wound healing assay. The results indicated that silencing FERMT2 remarkably inhibited the collective cell migration in SW1116 and SW1463 cells (Fig 6A-B). The cell migration ability detected by transwell assay or wound healing assay showed the same decreased trend in CRC cells after knockdown of FERMT2. In order to control the experimental error, we used transwell assay to detect the cell migration ability in the subsequent experiments.

Fig 6. Knockdown of FERMT2 inhibited the migration, invasion and EMT in SW1116 or SW1463 cells. (A-D) The migration ability of SW1116 or SW1463 cells after FERMT2 knockdown were detected by wound healing assay (A, B) or transwell assay (C, D). Scale bars: 100 μm. (E-H) The expression of E-cadherin, N-cadherin and Vimentin in SW1116 (E, F) or SW1463 (G, H) cells after FERMT2 knockdown were measured by western blot assay. All the tests were repeated at least three times. *P < 0.05, **P < 0.01vs shNC.

---

## [Editor Report · Decision Letter 1]

23 Nov 2022

Alzheimer’s risk factor FERMT2 promotes the progression of colorectal carcinoma via Wnt/β-catenin signaling pathway and contributes to the negative correlation between Alzheimer and cancer

PONE-D-22-25240R1

Dear Dr. Xu,

We’re pleased to inform you that your manuscript has been judged scientifically suitable for publication and will be formally accepted for publication once it meets all outstanding technical requirements.

Kind regards,

Frédéric André

Academic Editor

PLOS ONE
---

## [Editor Report · Acceptance letter]

28 Nov 2022

PONE-D-22-25240R1 

Alzheimer’s risk factor FERMT2 promotes the progression of colorectal carcinoma via Wnt/β-catenin signaling pathway and contributes to the negative correlation between Alzheimer and cancer 

Dear Dr. Xu:

I'm pleased to inform you that your manuscript has been deemed suitable for publication in PLOS ONE. Congratulations! Your manuscript is now with our production department. 

Kind regards, 

on behalf of

Dr. Frédéric André 

Academic Editor

PLOS ONE